# Biomarker Candidates for Alzheimer’s Disease Unraveled through In Silico Differential Gene Expression Analysis

**DOI:** 10.3390/diagnostics12051165

**Published:** 2022-05-07

**Authors:** Maria-del-Carmen Silva-Lucero, Jared Rivera-Osorio, Laura Gómez-Virgilio, Gustavo Lopez-Toledo, José Luna-Muñoz, Francisco Montiel-Sosa, Luis O. Soto-Rojas, Mar Pacheco-Herrero, Maria-del-Carmen Cardenas-Aguayo

**Affiliations:** 1Laboratory of Cellular Reprogramming, Departamento de Fisiología, Facultad de Medicina, Universidad Nacional Autónoma de México, Av. Universidad No. 3000, Coyoacán, Mexico City 04510, Mexico; carmenaguila10@hotmail.com (M.-d.-C.S.-L.); jaredrivera@ciencias.unam.mx (J.R.-O.); jalim166@gmail.com (L.G.-V.); gus_lt2@hotmail.com (G.L.-T.); 2Instituto de Neurobiología, Faculty of Psychology, National Autonomous University of Mexico, Av. Universidad 3004 Col Copilco-Universidad, Alcaldía, Coyoacán, Mexico City 04510, Mexico; 3Department of Physiology, Biophysics and Neurosciences, CINVESTAV-IPN, Mexico City 07360, Mexico; 4National Dementia BioBank, Ciencias Biológicas, Facultad de Estudios Superiores Cuautitlán, Universidad Nacional Autónoma de México, Cuautitlán Izcalli 53150, Mexico; jluna_tau67@comunidad.unam.mx (J.L.-M.); fmontiel_sosa@yahoo.com.mx (F.M.-S.); 5Banco Nacional de Cerebros-UNPHU, Universidad Nacional Pedro Henríquez Ureña, Santo Domingo 1423, Dominican Republic; 6Laboratorio de Patogenesis Molecular, Laboratorio 4, Edificio A4, Carrera Médico Cirujano, Facultad de Estudios Superiores Iztacala, Universidad Nacional Autónoma de México, Tlalnepantla de Baz 54090, Mexico; oskarsoto123@unam.mx; 7Neuroscience Research Laboratory, Faculty of Health Sciences, Pontificia Universidad Católica Madre y Maestra, Santiago de los Caballeros 51000, Dominican Republic; mpacheco@pucmm.edu.do

**Keywords:** Alzheimer’s disease, biomarkers, bioinformatics, differentially expressed genes

## Abstract

Alzheimer’s disease (AD) is neurodegeneration that accounts for 60–70% of dementia cases. Symptoms begin with mild memory difficulties and evolve towards cognitive impairment. The underlying risk factors remain primarily unclear for this heterogeneous disorder. Bioinformatics is a relevant research tool that allows for identifying several pathways related to AD. Open-access databases of RNA microarrays from the peripheral blood and brain of AD patients were analyzed after background correction and data normalization; the Limma package was used for differential expression analysis (DEA) through statistical R programming language. Data were corrected with the Benjamini and Hochberg approach, and genes with p-values equal to or less than 0.05 were considered to be significant. The direction of the change in gene expression was determined by its variation in the log2-fold change between healthy controls and patients. We performed the functional enrichment analysis of GO using goana and topGO-Limma. The functional enrichment analysis of DEGs showed upregulated (UR) pathways: behavior, nervous systems process, postsynapses, enzyme binding; downregulated (DR) were cellular component organization, RNA metabolic process, and signal transduction. Lastly, the intersection of DEGs in the three databases showed eight shared genes between brain and blood, with potential use as AD biomarkers for blood tests.

## 1. Introduction

Alzheimer’s disease (AD) is a complex and heterogeneous neurodegenerative disorder. AD is the most common cause of dementia, accounting for 60–80% of all these cases [1]. According to the World Alzheimer Report 2021, over 55 million people globally live with dementia [2,3,4,5,6], and this number is projected to increase to 152 million by 2050 [7,8]. An estimated 6.5 million Americans aged 65 and older are living with Alzheimer’s dementia in 2022, with 73% aged 75 or older [9]. Furthermore, the Alzheimer’s Association reported that Alzheimer’s deaths jumped 16% during the coronavirus pandemic (SARS-CoV-2).

AD is a progressive age-related neurodegenerative disorder. The most common type is late-onset or sporadic AD, defined as AD with an age of onset after age 65. It is attributed to a complex combination of genes, chemicals factors, environment, and lifestyle habits [10,11]. For instance, several studies suggested an association between AD and increased blood levels of toxic metals such as copper (Cu), selenium (Se), zinc (Zn), lead (Pb), and mercury [12,13]. Several investigations focused on elucidating the influence of these toxic metals and the molecular mechanisms involved in neurodegenerative disease development [14]. Elevated levels of nonessential metals may induce various detrimental intracellular events, including oxidative stress, mitochondrial dysfunction, DNA fragmentation, protein misfolding, endoplasmic reticulum stress, autophagy dysregulation, and the activation of apoptosis that may alter neurotransmission. Living in industrial areas constitutes an environmental hazard that could be an important contributing factor for the development of AD and other diseases because of exposure to high concentrations of heavy metals. Furthermore, there is increasing evidence suggesting the role of environmental factors in the development of AD, such as exposure to polychlorinated biphenyls (PCBs) and organochlorine pesticides (OC) [15,16]. The other type of AD is early-onset (EOAD) or familial AD, which occurs at ages of onset ranging from 30 to 50 years; it is hereditary and accounts for 1%–5% of AD cases; the remaining 90% are sporadic or late-onset AD (LOAD). EOAD involves mutations in genes encoding amyloid precursor protein (APP), presenilin-1 (PS1), and presenilin-2 (PS2). Mutations in these genes might result in the alteration of amyloid beta (Aβ) peptide production (both Aβ 40 and Aβ 42, and smaller Ab peptides), leading to cell death and dementia [17,18,19].

The development of intraneuronal and extracellular lesions at vulnerable sites in the brain is central to AD. Pathological hallmarks of AD are neuritic plaques and neurofibrillary tangles related to the accumulation of the amyloid-β (Aβ) peptide in brain tissue and to cytoskeletal abnormalities that are caused by the accumulation of hyperphosphorylated microtubule-associated protein tau in neurons, respectively [5,20,21]. The progression of AD pathology begins in structures of the entorhinal cortex (EC) and hippocampus (HIP) in the prodromal stage [22,23,24,25,26]. 

AD is a heterogeneous disease; novel approaches to integrating genetics, expression, and epigenetics into organized molecular networks may facilitate the understanding of the pathogenesis underlying these diseases [27]. Currently, AD does not have a cure, and confirmatory diagnostics is postmortem; however, some potential biomarkers in CSF were proposed for the diagnosis of AD, such as the presence of total tau protein (T-tau) and hyperphosphorylated tau (P-tau 181 and P-tau 217), the relationship between amyloid beta 42/40 peptides, and the presence of neurofilament light protein (NfL) [28,29,30]. The use of other diagnostic biomarkers such as YKL-40 associated with neuroinflammation was suggested [31]. More studies are needed to identify potential biomarkers of this disease for developing early diagnostics and new therapeutic approaches. Several high-throughput experimental approaches involving genomewide linkage (GWL) scans, genomewide association (GWA) studies, and genomewide expression (GWE) profiling were extensively utilized to identify the underlying genetic risk factors and new biomarkers [32,33,34]. Recent discoveries indicate that comprehensive bioinformatic analyses could allow for the discovery of new therapeutic targets in AD pathology [35,36]. In this study, we performed a bioinformatics analysis of open-access RNA microarrays from AD patients’ peripheral blood and brain to explore critical genes that might be involved in AD pathogenesis and could help in improving diagnosis. 

## 2. Materials and Methods

Figure 1 schematizes the bioinformatics analysis performed in this study. The down sections detail it to step by step.

### 2.1. Microarray Processing

In this work, all analyses were carried out in January 2022 with R version 4.1.2. First, we analyzed free-access microarray databases from the Gene Expression Omnibus (https://www.ncbi.nlm.nih.gov/geo/, accessed on 22 March 2022) from affected human blood (GSE4226, GSE63061, GSE85426, GSE97760, GSE140829, GSE18309) and brain areas (GSE132903, GSE118553, GSE110226, GSE84422, GSE28146, GSE33000, GSE48350, GSE4757, GSE39420) with sporadic Alzheimer disease. We only selected databases in which gene expression showed normal distribution in the histogram, similar a median across all samples in boxplot, and with AD samples and healthy samples grouped apart or with a tendency to separate in PCA plots. For further analysis, we worked with only female samples between 65 and 90 years old to reduce variability (Table 1).

Data from database GSE84422-GPL96 were preprocessed with the affy and Core R packages using the mas5 algorithm for background-noise correction and filtering absent probes. Subsequently, preprocessed data were normalized with quantile normalization and transformed into a base 2 logarithmic scale. For GSE97760 and GSE132903, we downloaded preprocessed data and normalized them with quantile normalization, and base 2 logarithmic scale transformation was applied to both. 

### 2.2. Differential Expression Analysis

We performed differential expression analysis (DEA) with processed data between the AD and CNTRL samples of each microarray database using the Limma R package. Genes with a logarithmic fold change (logFC) > 0.5 and a corrected p-value (adj.pval) < 0.05 (Benjamini–Hochberg for multiple tests) were considered to be upregulated differentially expressed genes (DEGs); genes with logFC < −0.5 and adj.pval. < 0.05 were considered to be downregulated DEGs. 

### 2.3. Functional Enrichment Analysis

In each database, we performed functional enrichment analysis (FEA) of up- and downregulated DEGs; we used the R topGO package with the weight01 algorithm. For statistical comparison, we carried out a Fisher’s exact test. We considered as background up and down DEGs. Lastly, we plotted the first 10 most enriched gene ontology terms (GO) for biological processes (BPs), cellular components (CCs), and molecular functions (MFs), the number of genes in each GO, and the significance of the enrichment. 

### 2.4. Genes in Common and Selection of Genes of Interest

To find the genes of interest, we looked for genes in common with the same pattern of expression in the blood and brain datasets. We compared DEGs found in blood with DEGs found in each brain database (GSE97760 ∩ GSE84422-GPL96; GSE97760 ∩ GSE132903). Lastly, we sought genes that were shared among all comparisons. 

## 3. Results

### 3.1. Homogenization of Raw Database Data

We selected and processed three open-access microarray databases to analyze transcriptomic differences in Alzheimer’s disease. These databases were obtained from blood samples (GSE97760, peripheral blood) and two from brain tissue (GSE84422-GPL96 of hippocamppus and GSE132903 of middle temporal gyrus). They are summarized in Table 1, they originated from patients with Alzheimer’s disease, and were compared with samples from control individuals.

Selected processed data showed normal distribution in their expression level and a similar median in all samples; this indicates that all three databases were suitable for further statistical analysis (Figure 2A–C). Subsequently, we performed principal component analysis (PCA) to observe the pattern between patient samples and controls from the processed data. Our results showed that the first and second principal components explained approximately 25% of variation in the transcriptomic profile. Subjects with Alzheimer’s disease were also grouped independently of control subjects, indicating that the two groups (patients vs. controls) differed in their transcriptional profiles (Figure 3A–C).

### 3.2. Differential Expression Analysis (DEA)

Differential expression analysis (DEA) was performed to evaluate the differences in the transcriptional profile between the groups of each database. Our results showed 850 downregulated and 693 upregulated DEGs for the blood database (Figure 4A) (GSE97760), and 17 downregulated and 81 upregulated DEGs in the brain database (GSE84422-GPL96) (Figure 4B). In the GSE132903 database, we found 167 downregulated DEGs and 156 upregulated DEGs (Figure 4C).

### 3.3. Functional Enrichment Analysis of DEGs

#### 3.3.1. Biological Process 

After finding many upregulated and downregulated DEGs, we determined in each database if these genes overrepresented any specific biological function, for which we performed functional enrichment analysis (FEA). For the blood database (GSE97760), we detected the enrichment of different mRNA biological processes: the regulation of transcription by RNA polymerase II and RNA processing were upregulated, while downregulated were cell development, G protein-coupled receptor signaling pathway, and animal organ morphogenesis (Figure 5A). Analysis for the brain database (GSE84422-GPL96) showed no upregulated biological processes, while downregulated processes were related to the RNA metabolic process (Figure 5B). Upregulated DEGs in the other brain database, GSE132903, were associated with cellular component assembly and the negative regulation of the apoptotic process. In contrast, those downregulated were associated with the signal transduction and establishment of localization in the cell (Figure 5C). 

#### 3.3.2. Molecular Function

In the case of molecular function, in the blood database (GSE97760), we found functions of metal ion binding, DNA binding, and RNA binding to be upregulated. Downregulated functions were organic cyclic compound binding, heterocyclic compound binding, and small-molecule binding (Figure 6A). In the brain database (GSE84422-GPL96), we found no upregulated molecular function, but found downregulated RNA binding function (Figure 6B). For database GSE132903, we found upregulated molecular functions to be DNA-binding transcription factor activity, RNA polymerase II cis-regulatory region sequence-specific DNA binding, and kinase activity, while downregulated were enzyme binding, catalytic activity (acting on a protein), and calcium ion binding (Figure 6C). 

#### 3.3.3. Cellular Component

Regarding enriched cellular components, we found that, for the blood database (GSE97760), upregulated enriched processes were the nucleus, nucleoplasm, and nucleolus. Downregulated were integral components of membrane, extracellular region, and extracellular space (Figure 7A). In the brain database (GSE84422-GPL96), we found the upregulation of cellular components related to neuron projection, axon, and postsynapse, and the protein-containing complex was downregulated (Figure 7B). In database GSE132903, we found upregulated alterations for cellular components such as chromatin, and intracellular organelle lumen, synapse, endoplasmic reticulum, and membrane protein complex were downregulated (Figure 7C).

### 3.4. Genes of Great Interest as Potential AD Biomarkers

We looked for the intersection of genes between the two brain databases and the blood database, which are of great interest as potential AD biomarkers for blood tests. We found genes of interest in common (shared between the blood and brain databases) with the same expression pattern (up- or downregulated) between the blood and brain datasets. As mentioned in Section 2, the blood database (GSE97760) was analyzed with the two brain databases (GSE84422-GPL96 and GSE132903). In the first intersection between the blood database (GSE97760) and one of the brain databases (GSE84422-GPL96), we found three genes of interest, namely, PPP3CB, SNCB, and SACS, and all were upregulated (Figure 8A). Five genes of interest were found for the intersection between the blood database (GSE97760) and brain database GSE132903: SNCA, FKBP1B, JMY, ZNF525, and COBLL1 (Figure 8B). 

## 4. Discussion

The analysis of the gene expression dataset and the identification of differentially expressed genes in disease compared to a healthy condition target different nodes for novel biomarker detection. 

Regarding AD, previous studies showed that abnormalities in the pre- and/or postsynaptic machinery are compromised in many age-related neurological disorders, including AD [37,38,39]. Upon subjection to enrichment analysis, our DEGs revealed that they are involved in diverse processes that could be related to this neurodegenerative disease. In this sense, some enriched GO biological processes are relevant to disease due their relationship with the synaptic process. Analysis of the blood database (GSE97760) showed that downregulated DEGs are associated with axon guidance and behavior, while the brain database (GSE84422) showed downregulated DEGs associated with learning or memory. In the case of molecular function, in the blood database (GSE97760), we found downregulated molecular function of ion channel activity, calcium ion binding, and gated channel activity. In contrast, the brain database (GSE84422) showed downregulated DEGs associated with calcium ion binding. Lastly, upregulated cellular components in the brain database (GSE84422-GPL96) were related to neuron projection, axon, and postsynapse. A previous study demonstrated the association between miRNAs in AD brains and their target genes using bioinformatics analysis; findings showed dysregulated miRNAs and their target mRNAs involved in biological processes such as postsynaptic machinery, neurotransmission, and neuronal viability in AD [40]. Lastly, the results of an approach to identify common signature patterns across public AD studies suggest that AD gene regulatory networks (GRNs) show significant enrichment for key signaling mechanisms involved in neurotransmission. Prioritized genes were prominent in synaptic transmission and implicated in cognitive deficits [41].

The results of the present bioinformatic analysis to explore critical genes potentially involved in the pathogenesis of AD revealed eight interesting genes, which are described below.

The first gene to emerge from the intersection analysis of open-access microarray databases GSE97760 (peripheral blood) and GSE84422-GPL96 (hippocampus) was calcineurin A subunit β (PPP3CB), which is upregulated.

Calcineurin is a major calmodulin-binding protein in the brain and the only serine/threonine phosphatase under the control of Ca^2+^/calmodulin. Calcineurin is always present as a heterodimer, and it consists of a 58–64 kDa catalytic subunit, calcineurin A, and a 19 kDa regulatory subunit, calcineurin B [42]. The calcineurin A subunit is differentiated into three types, namely, Aα, Aβ (PPP3CB), and Aγ, derived from three different genes. Calcineurin Aα and Aβ are highly expressed in the brain, whereas calcineurin Aγ expression is specific for the testis [43].

A study showed that calcineurin Aβ is upregulated in the hippocampus in the early stages of AD, and these are data correlated with our finding of the upregulation of calcineurin A subunit β in both the blood and brain databases. In that study, the authors compared gene expression in the hippocampus, a region relatively susceptible to neurodegeneration by neurofibrillary tangles (NFTs), with its expression in the parietal cortex, a brain region more resistant to this type of degeneration [44].

At the molecular level, calcineurin may be involved in the regulation of tau phosphorylation in AD brains [45,46]. Another possible explanation of the role of calcineurin Aβ in AD is that calcineurin may be involved in neuronal cell death triggered by insults that increase cytosolic Ca^2+^ [47] or that the overexpression of calcineurin triggers cytochrome c/caspase-3-dependent apoptosis in neurons [48]. Lastly, one study concluded that the increased expression of calcineurin Aβ might alter APP metabolism and lead to increased production of amyloid Aβ, a major cause of AD. Therefore, the downregulation of calcineurin Aβ levels could be used as a potential therapeutic agent to reduce amyloid Aβ levels [49].

The second gene obtained in our study is beta-synuclein (SNCB), which is upregulated. Beta-synuclein (βSyn) is a presynaptic protein that is expressed in the central nervous system (CNS) and highly enriched in the hippocampus [50]. A significant advantage of βSyn compared with other synaptic CSF markers, such as neurogranin, is its specific expression in the CNS, which is why the release of βSyn from degenerating neurons is more likely to also be detected in blood [51].

The physiological function of βSyn is unclear, and it was studied in the context of α-synucleinopathies such as Parkinson’s disease (PD) and dementia with Lewy bodies (DLB) [52,53]. No significant changes in βSyn levels were observed in CSF and serum from PD patients. Therefore, it is unlikely that changes in βSyn in AD are due to α-synuclein copathology. Because the pattern of βSyn changes in CSF and serum resembles other synaptic proteins in neurodegenerative diseases, total βSyn levels reflect synaptic degeneration rather than specific changes associated with βSyn pathology [51].

The establishment of a new detection method for beta-synuclein provides evidence that beta-synuclein is a novel diagnostic and predictive biomarker candidate for AD when measured in CSF. The CSF levels of presynaptic beta-synuclein could be used as a marker of synaptic degeneration and may thus be suitable as a measurement in clinical trials targeting synapse loss AD [54].

The third gene, a product of intersection analysis, is sacsin (SACS), which is a cochaperone that acts as a regulator of Hsp70 chaperone machinery and may be involved in the processing of other ataxia-linked proteins. SACS was upregulated in AD tissue. SACS encodes the sacsin protein, which contains a UbL domain at the N terminus, a DnaJ domain, and a HEPN domain at the C terminus. The gene is highly expressed in the central nervous system, and is also found in the skin, skeletal muscle, and in small amounts in the pancreas [55]. 

Sacsin is predominantly localized in the cytoplasm and in mitochondria. Sacsin interacts with dynamin-related protein 1 (DRP1), a GTPase that mediates mitochondrial fission. In a sacsin knockout mouse, mitochondria appeared to be overly fused and showed reduction in mobility [56]. These data suggest that sacsin may also participate in mitochondrial fission. Mitochondrial dysfunction is a common feature in many neurodegenerative diseases such as Alzheimer’s [57].

The first gene to emerge from our intersection analysis of open-access microarray databases GSE97760 (peripheral blood) and GSE132903 (middle temporal gyrus) was alpha-synuclein (SNCA), which is downregulated.

α-Synuclein (αSyn) is predominantly expressed within presynaptic terminals of neurons [58]. The expression of αSyn can mainly be found in the cerebral cortex, cerebellum, striatum, thalamus, hippocampus, and olfactory bulb [59]. α-Synuclein is a small 140-residue protein translated from 5 SNCA gene exons located on the long arm of chromosome 4 [60]. The primary structure of αSyn is subdivided into three regions: an amphipathic N-terminal, a hydrophobic core region known as the NAC, and an unstructured acidic C-terminal.

In addition to AD brain pathology, including Aβ and tau lesions, studies documented the occurrence of comorbid αSyn or Lewy-related pathology (LRP) in more than 50% of autopsy-confirmed AD brains [61]. The co-occurrence of LRP in AD is associated with the immunohistochemical colocalization of αSyn and tau pathology [62], and to a lesser extent to αSyn and Aβ pathology [63].

A study conducted with CSF samples collected over a period of 7 years as part of the Alzheimer’s Disease Neuroimaging Initiative (ADNI) showed that lower α-syn/p-tau181 levels were associated with both the faster progression of cognitive decline and the conversion of MCI into AD [64]. The results of another study showed lower levels of α-syn and its heterocomplexes (i.e., α-syn/Aβ and α-syn/tau) in red blood cells (RBCs) of AD patients compared with healthy controls (HCTLS). Both α-syn/Aβ and α-syn/tau heterodimers in RBCs distinguished AD participants from healthy controls with sufficient accuracy. These data suggest that α-syn heteroaggregates in erythrocytes are a potential tool for the early diagnosis of AD on blood tests [65].

The second obtained gene is FKBP prolyl isomerase 1B (FKBP1B), which is downregulated. 

FK506-binding proteins 1a and 1b (FKBP1a/1b) are immunophilin proteins that bind immunosuppressive drugs FK506 and rapamycin. Many immunophilins exhibit peptidyl-prolyl isomerase activity, and function as protein chaperones and structural stabilizers [66,67]. Microarray studies revealed that FKBP1b gene expression was downregulated in the hippocampus of aging rats and in early Alzheimer’s disease patients. These results suggest that declining FKBP function is a key factor in age-related Ca^2+^ dysregulation in the brain [68]. Furthermore, one paper reported gene therapy approaches, and found that increasing FKBP1b reversed calcium dysregulation and memory impairment in aging rats, allowing for them to perform as well as young rats on a memory task [69].

Other genes in analysis (JMY, ZNF525, and COBLL1) were not previously associated with Alzheimer’s disease. However, they could be considered in future studies. JMY is a gene that encodes for junction-mediating and regulatory protein, and it acts as both a nuclear p53/TP53-cofactor and a cytoplasmic regulator of actin dynamics depending on conditions [70]. During autophagy, actin filament networks move and remodel cellular membranes to form autophagosomes that enclose and metabolize cytoplasmic contents. LC3 and STRAP regulate JMY’s actin assembly activities in trans during autophagy [70]. Thus, the connection of JYM to AD may be at the level of autophagy regulation, since autophagy is impaired in AD neurons and other peripheral cells. ZNF525 encodes for zinc finger protein 525, which may be involved in transcriptional regulation. Sterner et al. (2013) reported that the expression of ZNF525 in the cerebral cortex broadly mirrors developmental patterns of cortical glucose consumption [71]. Therefore, this protein could be related to brain metabolic function, which is affected in AD. In turn, Cordon-Bleu WH2 repeat-protein-like 1 (COBLL1) enables cadherin binding activity and it is located in extracellular exosomes. An important paralog of this gene is COBL, which is involved in neural tube formation [72,73]. Gene Ontology (GO) annotations related to this gene include actin binding. Diseases associated with COBLL1 include macular degeneration age-related 10 (ARMD10). An association among COBLL1 rs7607980 C allele, lower serum insulin levels, and lower insulin resistance was reported in overweight and obese children. Hyperglycemia is a risk factor for dementia [74,75].

Bioinformatics analysis has been applied to many diseases in recent decades in the search for identifying new biomarkers for disease diagnosis and treatment [76,77,78].

DEGs in AD were also studied. Results in Gene Ontology (GO) terms of those articles are similar to ours [40,79]. Our results are reliable for the following reasons: (1) This study follows statistical test models along with gene enrichment to better understand detailed molecular mechanisms underlying AD and thus assist in the selection of possible biomarkers for patients with the disease. (2) The three selected datasets are all based on the same platform. They could be combined into one dataset for normalized analysis, which would improve the credibility of the results. (3) During investigation, the original data were processed to ensure that the information was valid and consistent. 

However, there were a few limitations in our study. The sample size for microarray analysis was small (we selected three datasets with a sample size of 58), which may have caused a relatively high rate of false-positive results. Moreover, no experimental verification was performed. Thus, to verify the current results, further studies are warranted to elucidate the biological function of these genes in AD besides their potential use as reliable molecular biomarkers for disease.

In summary, eight genes involved in AD pathogenesis identified in our bioinformatic study shared between the blood database and the two brain tissue databases are potential biomarkers for AD. More research is warranted in order to validate its detection in patients’ peripheral blood samples or CSF. Among these presumptive AD biomarkers are calcineurin subunit β, a serine/threonine phosphatase under the control of Ca^2+^/calmodulin; β-synuclein, which is associated with synaptic degeneration; α-synuclein, which is associated with the occurrence of comorbid LRP in AD; and FKBP prolyl isomerase 1B (FKBP1B), a chaperone involved in age-related Ca2+ dysregulation in the brain. The SACS gene may be another candidate molecule for AD diagnosis because of its involvement in mitochondrial dysfunction, which is part of the pathophysiology of AD, and three genes JMY, ZNF525, and COBLL1, which are involved in the generation of autophagosomes, transcription regulation glucose consumption, and cadherin/actin binding linked to lower serum insulin levels, respectively, could also be useful in AD diagnosis. 

## 5. Conclusions

The differential gene expression analysis of tissue (hippocampus and middle temporal gyrus) and peripheral blood databases from Alzheimer’s disease patients resulted in the functional enrichment of DEGs showing UR pathways of behavior, nervous system processes, postsynapses, and enzyme binding, while cellular component organization, RNA metabolic process, and signal transduction were DR. Lastly, the intersection of DEGs in the three databases revealed eight common genes in the brain and blood that could be validated in samples from different ethnic groups, such as Mexican AD patients, as potential new biomarkers with possible application for AD diagnostic blood tests.

## Figures and Tables

**Figure 1 diagnostics-12-01165-f001:**
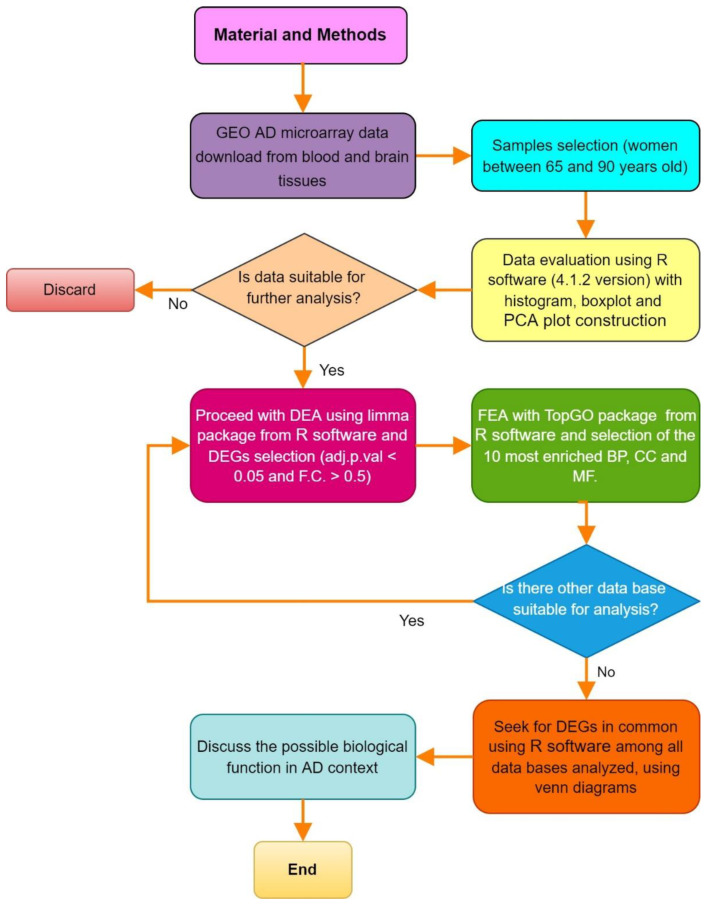
Flowchart of bioinformatics analysis performed in this study with detailed software, database, and tools used for each step (AD, Alzheimer´s disease; GEO, Gene Expression Omnibus; PCA, principal component analysis; DEA, differential expression analysis; DEGs, differentially expressed genes; FEA, functional enrichment analysis; GO, gene ontology; BP, biological processes; CC, cellular component; MF, molecular function).

**Figure 2 diagnostics-12-01165-f002:**
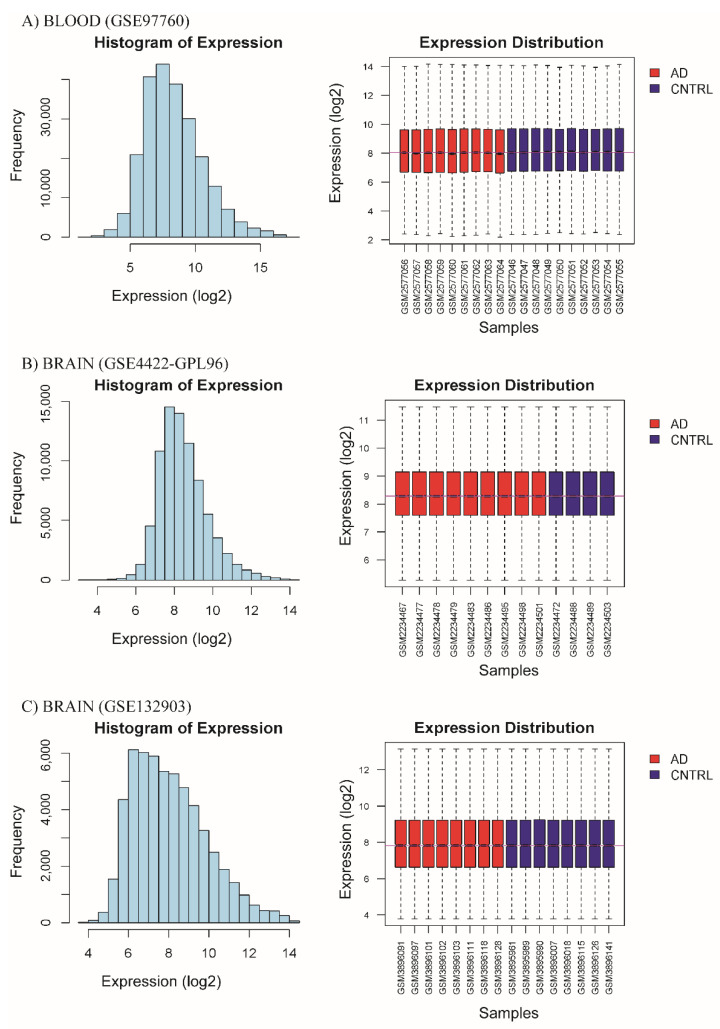
Homogenization of data, distribution histogram of logarithmic-scale expression after processing data, and Tukey diagrams with expression value distribution in a logarithmic scale of each subject in both groups (CNTL and AD). Data of (**A**) blood tissue from database GSE97760, (**B**) brain tissue from database GSE84422-GPL96, and (**C**) brain tissue from database GSE132903.

**Figure 3 diagnostics-12-01165-f003:**

Principal component analysis (PCA) of processed data (CNTL and AD). Data of (**A**) blood database GSE97760, (**B**) brain database GSE84422-GPL96, and (**C**) brain database GSE132903.

**Figure 4 diagnostics-12-01165-f004:**
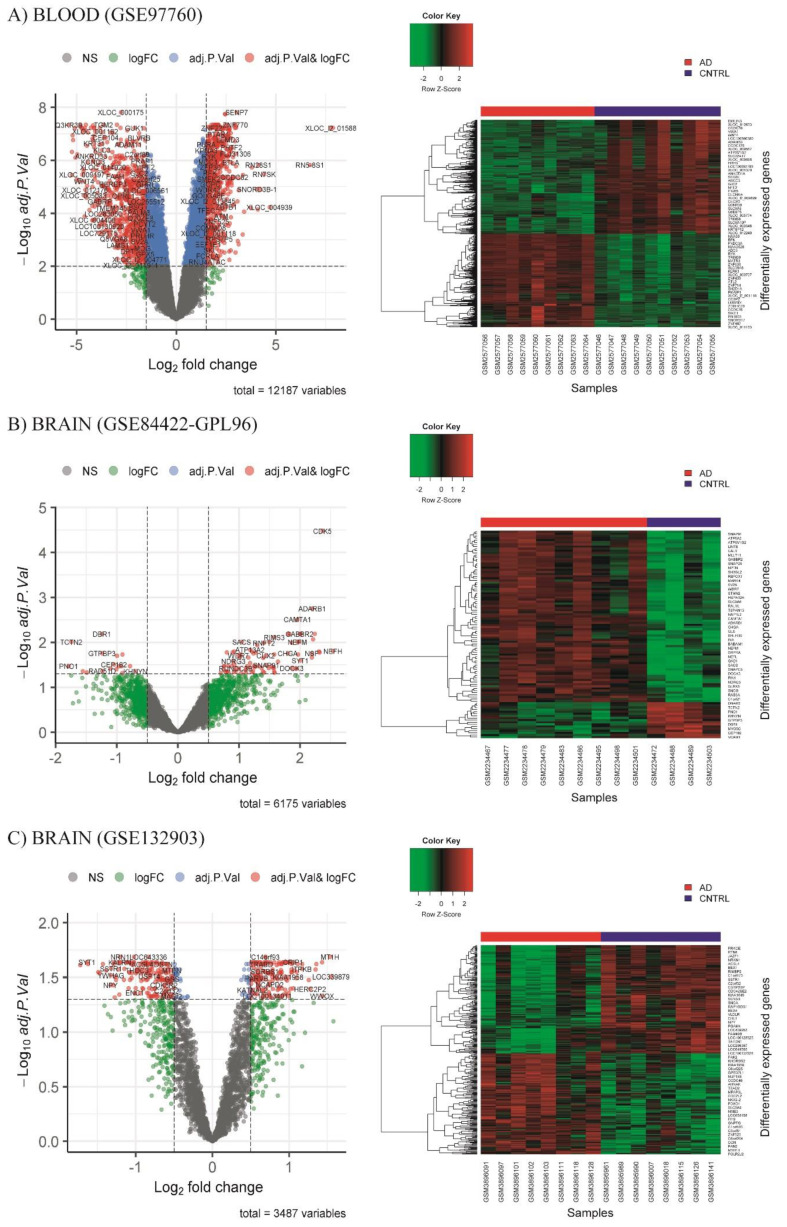
Volcano plot of up- and downregulated differentially expressed genes (DEGs) when comparing Alzheimer’s disease (AD) samples with healthy control samples (CNTRL), and heat map with logarithmic scale expression of DEGs in AD and CNTRL samples. CNTRL in blue, AD in red, purple line = median value, principal component 1, Dim1; principal component 2, Dim2; not significant, NS; genes with −0.5 > fold change < 0.5 (green dots); genes with −0.5 > fold change < 0.5 and adjusted p value (adj.P.val) < 0.05 (red dots). Data of (**A**) blood GSE97760, (**B**) brain GSE84422-GPL96, and (**C**) brain GSE132903.

**Figure 5 diagnostics-12-01165-f005:**
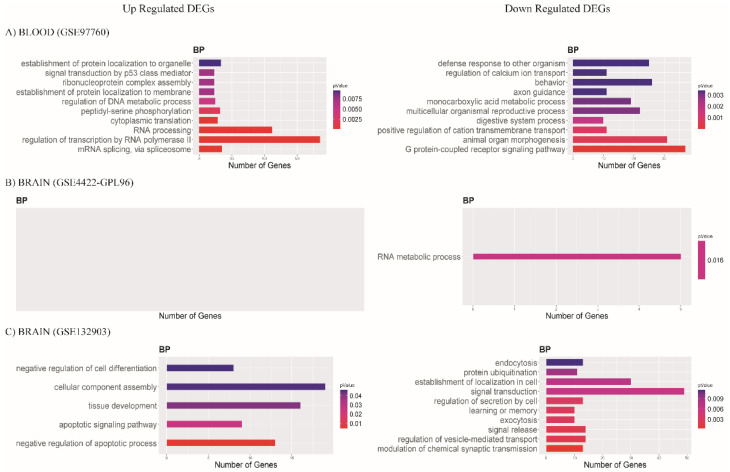
Bar plots show biological process (BP) and most enriched GO terms with *p*-value < 0.05. Color scale represents p-value; X axis indicates number of genes in each GO term. Data of (**A**) blood GSE97760, (**B**) brain database GSE84422-GPL96, and (**C**) brain database GSE132903.

**Figure 6 diagnostics-12-01165-f006:**
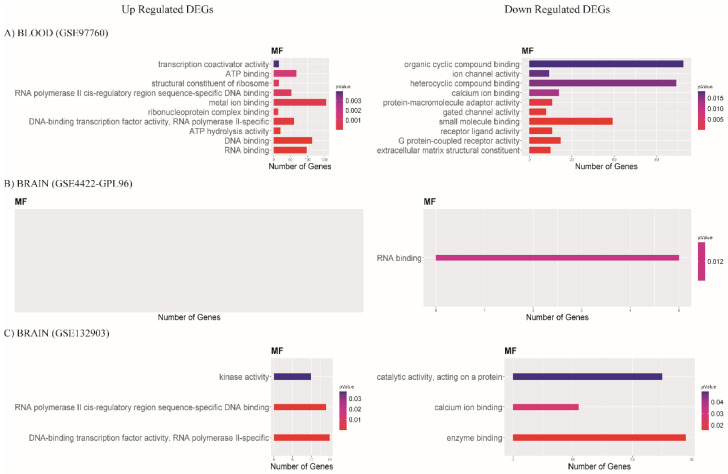
Bar plots show molecular functions (MFs); most enriched GO terms with *p*-value < 0.05. Color scale represents p-value; X axis shows the number of genes in each GO term. Data of (**A**) blood database GSE97760, (**B**) brain database GSE84422-GPL96, and (**C**) brain database GSE132903.

**Figure 7 diagnostics-12-01165-f007:**
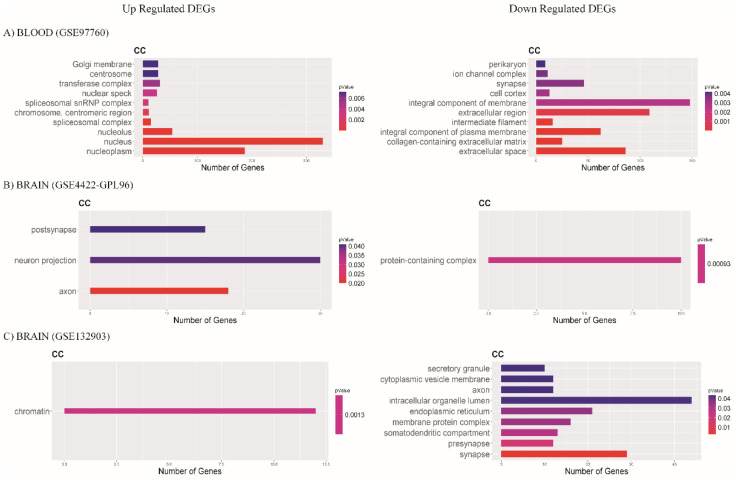
Bar plots show cell components (CC); most enriched GO terms with *p*-value < 0.05. Color scale represents p-value; X axis shows the number of genes in each GO term. Data of (**A**) blood database GSE97760, (**B**) brain database GSE84422-GPL96, and (**C**) brain database GSE132903.

**Figure 8 diagnostics-12-01165-f008:**
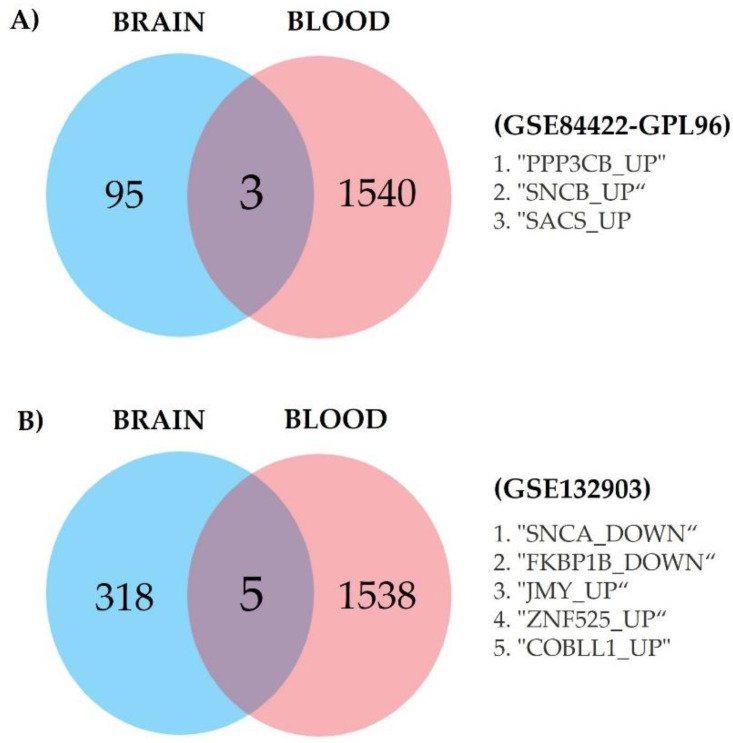
Venn diagram of overlap between genes from differential expression analysis and weighted gene correlation network analysis for AD phenotype (DEGs+WGCNA), previously reported differentially expressed genes, and list of mapped genes obtained from genomewide association studies (AD GWAS); intersection (∩). (**A**) GSE97760 ∩ GSE84422-GPL96; (**B**) GSE97760 ∩ GSE132903.

**Table 1 diagnostics-12-01165-t001:** Selected database characteristics.

GEO ID	Tissue	Samples	Condition	Race	Publication and Last Update Date
GSE97760	Peripheral blood	CNTRL: 10 AD: 9	CNTRL: healthy AD: sporadic/advanced	CNTRL: not provided AD: not provided	Published on 14 April 2017/updated on 27 March 2018
GSE84422-GPL96	Hippocampus	CNTRL: 4 AD: 9	CNTRL: healthy AD: sporadic/definite	CNTRL: 4 white AD: 8 white/1black	Published on 19 August 2016/updated on 26 June 2019
GSE132903	Middle temporal gyrus	CNTRL: 8 AD: 8	CNTRL: healthy AD: sporadic/advanced	CNTRL: not provided AD: not provided	Updated on 18 September 2019

## Data Availability

All analyzed databases are open-access from GEO2R (See Table 1, https://www.ncbi.nlm.nih.gov/geo/, accessed on 22 March 2022).

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
