# Peer review of "Biomarker Candidates for Alzheimer’s Disease Unraveled through In Silico Differential Gene Expression Analysis"

_diagnostics, 2022, doi:10.3390/diagnostics12051165_

Round 1

Reviewer 1 Report

The scientific paper by Silva-Lucero et al. entitled “In Silico differential gene expression analysis in tissue data- 2 bases from patients with Alzheimer´s Disease, to identify potential new biomarkers.” aimed to perform a bioinformatics analysis to explore critical genes that might potentially be involved in AD's pathogenesis and could help improve the diagnosis. The methodology used in this research is appropriate for the hypotheses tested and the conclusions discussed are consistent with experimental data. However, the manuscript would benefit from some corrections and clarifications.

The title is too long and could be shortened. “In Silico differential gene expression analysis to identify potential new biomarkers for Alzheimer´s Disease” should be enough for readers to understand the main purpose of the manuscript.

Introduction section is too short and could be broadened to include more data on various factors which might be connected with the development of AD. The authors have mentioned environmental factors, but not chemicals (especially toxic metals, pesticides, PCBs, etc.), which have largely been connected with development of this disease in the recent years. Examples of recent studies covering this topic include: https://doi.org/10.1016/j.fct.2022.112839, https://doi.org/10.1016/j.envres.2021.110727, https://doi.org/10.1007/s11033-021-06386-x, https://doi.org/10.1186/s12940-019-0494-2

In the M&M section, approximate date of in silico analysis should be included having in mind that databases and bioinformatics tools (Gene Expression Omnibus) are constantly updated. Furthermore, the manuscript would benefit from adding a flow chart including detailed depiction of the analysis (which software/database/tool was used for each step of the analysis).

In the results section, the authors mentioned that they downloaded and processed 4 open access microarray analysis databases. Selection criteria for the databases should be explained in more detail in M&M section.

The authors should provide larger figures of better quality (especially in the case of Fig. 4, 5 and 6). Text on the figures is too small and almost impossible to read.

Although all the conclusions are sound, it should be noted that bioinformatics approach, as such, can only be viewed as a prioritization method for further laboratory testing. In the discussion section, it is necessary to explain all pros and cons of the used approach, with special reference to its limitations. For example, limitations of in silico data mining are thoroughly explained in the following papers: https://doi.org/10.32604/biocell.2022.018271, https://doi.org/10.1016/j.chemosphere.2020.128362,  https://doi.org/10.1016/j.biopha.2021.112598.

Author Response

Response to Reviewer 1

We sincerely appreciate your kind and important suggestions. Here are the point-by-point answers to your requests.

Point 1: “The title is too long and could be shortened. “In Silico differential gene expression analysis to identify potential new biomarkers for Alzheimer´s Disease” should be enough for readers to understand the main purpose of the manuscript”.

Response 1: We consider this suggestion is very useful and we have changed the title as follows (Lines 2 and 3):

“Biomarker candidates for Alzheimer’s disease unraveled through in silico differential gene expression analysis”.

Point 2: “Introduction section is too short and could be broadened to include more data on various factors which might be connected with the development of AD. The authors have mentioned environmental factors, but not chemicals (especially toxic metals, pesticides, PCBs, etc.), which have largely been connected with development of this disease in the recent years. Examples of recent studies covering this topic include: https://doi.org/10.1016/j.fct.2022.112839, https://doi.org/10.1016/j.envres.2021.110727, https://doi.org/10.1007/s11033-021-06386-x, https://doi.org/10.1186/s12940-019-0494-2

Response 2: We consider this suggestion is very useful and we have incorporated more information about recent statistics of AD and chemical factors connected with development of this disease. Please see lines: 48 – 78. We also improved all the redaction and spelling of our maniuscrip.

Point 3: “In the M&M section, approximate date of in silico analysis should be included having in mind that databases and bioinformatics tools (Gene Expression Omnibus) are constantly updated. Furthermore, the manuscript would benefit from adding a flow chart including detailed depiction of the analysis (which software/database/tool was used for each step of the analysis).”

Response 3: We have considered this suggestion and added the date of in silico analysis (Line 130-141). Besides, a new figure (Figure 1), which includes a flow chart with a detailed depiction of the analysis, was added. (Line 105-128).

Point 4: “In the results section, the authors mentioned that they downloaded and processed 4 open access microarray analysis databases. Selection criteria for the databases should be explained in more detail in M&M section.”

Response 4: We have included your suggested information in M&M section. Please see lines: 130-139.

Point 5: “The authors should provide larger figures of better quality (especially in the case of Fig. 4, 5 and 6). Text on the figures is too small and almost impossible to read.”

Response 5: We have corrected the quality, enlarged the text in all the figures, and updated their numbers because we now have an extra figure. Please see lines: 186, 192, 226, 249, 265, 280 and 298.

Point 6: “Although all the conclusions are sound, it should be noted that bioinformatics approach, as such, can only be viewed as a prioritization method for further laboratory testing. In the discussion section, it is necessary to explain all pros and cons of the used approach, with special reference to its limitations. For example, limitations of in silico data mining are thoroughly explained in the following papers:

https://doi.org/10.32604/biocell.2022.018271

https://doi.org/10.1016/j.chemosphere.2020.128362,  https://doi.org/10.1016/j.biopha.2021.112598.”

Response 6: This is a relevant suggestion. We have included the following paragraphs between lines 446-262:

Bioinformatics analysis has been applied to many diseases in recent decades in the search for identifying new biomarkers for disease diagnosis and treatment [1-3].

It is worth mentioning that there have been articles that study DEGs in AD. The results in Gene Ontology (GO) terms of those articles are similar to ours [4,5]. Besides, our results are reliable for the following reasons: (1) This study follows the statistical test models along with genes enrichment to better understand the detailed molecular mechanisms underlying AD and, thus, assist in the selection of possible biomarkers for patients with the disease. (2) The selected three datasets are all based on the same platform. They could combine into one dataset for normalized analysis, which would increase the credibility of the results. (3) During the investigation, the original data were processed to ensure that the information was valid and consistent.

However, there were a few limitations in our study. The sample size for microarrays analysis was small (we selected three datasets with a sample size of 58), which may cause a relatively high rate of false-positive results. Moreover, no experimental verification has been performed jet. Thus, to verify the current results, further studies are warranted to elucidate the biological function of these genes in AD besides their potential use as reliable molecular biomarkers for disease.

Final note:

We highlighted the new changes in green in the revised version.

References

  1. Sun, Y.; Chen, G.; Liu, Z.; Yu, L.; Shang, Y. A bioinformatics analysis to identify novel biomarkers for prognosis of pulmonary tuberculosis. BMC Pulm Med 2020, 20, 279, doi:10.1186/s12890-020-01316-2.
  2. Zhao, X.M.; Li, Y.B.; Sun, P.; Pu, Y.D.; Shan, M.J.; Zhang, Y.M. Bioinformatics analysis of key biomarkers for retinoblastoma. J Int Med Res 2021, 49, 3000605211022210, doi:10.1177/03000605211022210.
  3. Wei, D.; Li, R.; Si, T.; He, H.; Wu, W. Screening and bioinformatics analysis of key biomarkers in acute myocardial infarction. Pteridines 2021, 32, 79-92, doi:doi:10.1515/pteridines-2020-0031.
  4. Ceylan, H. Integrated Bioinformatics Analysis to Identify Alternative Therapeutic Targets for Alzheimer's Disease: Insights from a Synaptic Machinery Perspective. J Mol Neurosci 2022, 72, 273-286, doi:10.1007/s12031-021-01893-9.
  5. Yu, W.; Yu, W.; Yang, Y.; Lu, Y. Exploring the Key Genes and Identification of Potential Diagnosis Biomarkers in Alzheimer's Disease Using Bioinformatics Analysis. Front Aging Neurosci 2021, 13, 602781, doi:10.3389/fnagi.2021.602781.

Reviewer 2 Report

Alzheimer's disease (AD) is a progressive neurodegenerative disease. The dementia symptoms gradually worsen over several years. It is critical to identify the most suitable biomarkers to facilitate the appropriate diagnosis and treatment of AD. In this study, a differential gene expression analysis, using blood and two brain databases, was carried out to unravel biomarker candidates for AD. It was a well-designed study, and the manuscript is interesting however I would like to suggest some alterations to improve the quality of the manuscript. Therefore, authors may find below a list of points to be reviewed and changed.

  1. Title – In my opinion, the title is too extensive and a bit confusing. I would suggest a title like this: “Biomarker candidates for Alzheimer’s disease unraveled through in silico differential gene expression analysis”.
  2. I suggest revising the English throughout the manuscript, some minor errors can be found.
  3. Introduction
    - Line 48, please update the number of dementia cases worldwide for 2020.
    - Line 61, I believe references 8 and 9 are not the most appropriate. Please change the citations.
    - Line 67, AD definitive diagnosis can only be made postmortem, however, the biomarker triplet (Abeta, Total-Tau and P-Tau 181) in CSF is a valuable tool for AD diagnosis. Please include this topic in the introduction.
  4. Methods
    - Table 1 – Please indicate databases publication and last update dates.
    - Line 89 – The authors said: “normalization and/or base 2 logarithmic scale transformation was applied when needed”. Can the authors clarify this?
  5. Results
    - Figure 1 – In Tukey diagrams, if possible, move “CNTRL” and “AD” out of the graphs. Samples identifications are very small, please increase the figure or at least the letters size.
    - Figure 3 – I understand that it would be difficult to read all DEGs depicted, however, I believe that the size image can be increased. The caption of Figure 3A (caption CNTRL and AD) is not consistent with 3B and 3C (AD and CNTRL). Please change this.
    - Line 161 – The authors said: “…while those down-regulated were: cell development and G protein-coupled receptor signaling pathway”. I could not find the cell development term in the figure. Please clarify.
    - Lines 183- 184 – The authors said “catalytic activity, and acting on a protein”. Please correct this for “catalytic activity (acting on a protein) and calcium ion binding”.
    - Figures 4, 5 and 6 – The bars could have the same width and the GO terms should have the same size from A to C. Please improve image resolution if possible.
    - Line 211 – Correct for “Figure 7A”.
  6. Discussion – This topic is already extensive; however, I believe it would be interesting to discuss the significantly enriched Gene Ontology terms found for the databases. Are some of these terms most relevant to AD?
  7. Finally, I would like to ask why the authors choose the databases included in the manuscript and not others? What turns these more attractive? I noticed that these databases come from studies with small sample groups.

Author Response

Response to Reviewer 2

We sincerely appreciate your kind and important suggestions. Here are the point-by-point answers to your requests.

Point 1: In my opinion, the title is too extensive and a bit confusing. I would suggest a title like this: “Biomarker candidates for Alzheimer’s disease unraveled through in silico differential gene expression analysis”.

Response 1: We consider this suggestion is very useful and we have changed the title as follows (Lines 2 and 3):

“Biomarker candidates for Alzheimer’s disease unraveled through in silico differential gene expression analysis”.

Point 2: Introduction

- Line 48, please update the number of dementia cases worldwide for 2020.

- Line 61, I believe references 8 and 9 are not the most appropriate. Please change the citations.

- Line 67, AD definitive diagnosis can only be made postmortem, however, the biomarker triplet (Abeta, Total-Tau and P-Tau 181) in CSF is a valuable tool for AD diagnosis. Please include this topic in the introduction.

Response 2: We have did the changes suggested in the introduction. Please see the lines:

  • Lines 50 -51: update the number of dementia cases worldwide for 2020
  • Lines 50 -72: References changed
  • Lines 88 – 94: Topic about biomarkers in CSF included.

Point 3: Methods

- Table 1 – Please indicate databases publication and last update dates.

- Line 89 – The authors said: “normalization and/or base 2 logarithmic scale transformation was applied when needed”. Can the authors clarify this?:

Response 3: We thank the reviewer for the comments. Please see the changes in the next lines:

  • Lines 142-147: We added information and a column in table 1 with databases publication and last update dates.
  • Lines 113 - 115: We clarify the line indicated.

Point 4: Results

- Figure 1 – In Tukey diagrams, if possible, move “CNTRL” and “AD” out of the graphs. Samples identifications are very small, please increase the figure or at least the letters size.

- Figure 3 – I understand that it would be difficult to read all DEGs depicted, however, I believe that the size image can be increased. The caption of Figure 3A (caption CNTRL and AD) is not consistent with 3B and 3C (AD and CNTRL). Please change this.

- Line 161 – The authors said: “…while those down-regulated were: cell development and G protein-coupled receptor signaling pathway”. I could not find the cell development term in the figure. Please clarify.

- Lines 183- 184 – The authors said “catalytic activity, and acting on a protein”. Please correct this for “catalytic activity (acting on a protein) and calcium ion binding”.

- Figures 4, 5 and 6 – The bars could have the same width and the GO terms should have the same size from A to C. Please improve image resolution if possible.

- Line 211 – Correct for “Figure 7A”.

Response 4:  We have included the changes suggested. Please see the next lines:

  • Line 187: Figure 2 (before 1) was modified.
  • Line 227: Figure 4 (before 3) was changed.
  • Lines 238- 242: We clarify the line indicated.
  • Lines 264: We corrected the words indicated.
  • Lines 250, 266 and 281: Figures were modified as requested and improved.
  • Line 298: Correction realized.

Point 5: “Discussion – This topic is already extensive; however, I believe it would be interesting to discuss the significantly enriched Gene Ontology terms found for the databases. Are some of these terms most relevant to AD?”

Response 5:

This is a relevant suggestion. We have included the following paragraphs between lines 305-329:

Analysis of gene expression dataset and identification of differentially expressed genes in disease compared to a healthy condition is a way of targeting different nodes for the novel biomarkers detection.

Regarding AD, previous studies have shown that abnormalities in the pre-and/or postsynaptic machinery are compromised in many age-related neurological disorders, including AD [1-3]. Upon subjection to enrichment analysis, our DEGs revealed that they are involved in diverse processes that could be related to this neurodegenerative disease. In this sense, some enriched GO biological processes are relevant to disease due their relationship with the synaptic process. The analysis of the blood database (GSE97760) shows that downregulated DEGs are associated with axon guidance and behavior, whilst the brain database (GSE84422) shows downregulated DEGs associated with learning or memory. In the case of molecular function, in the blood database (GSE97760), we found downregulated molecular function of ion channel activity, calcium ion binding, and gated channel activity. In contrast, the brain database (GSE84422) shows downregulated DEGs associated with calcium ion binding. Finally, we found that up-regulated cellular components in brain database (GSE84422-GPL96) are related to neuron projection, axon, and post synapse. On the other hand, a previous study demonstrated the association between miRNAs in AD brains and their target genes using bioinformatics analysis, the findings showed dysregulated miRNAs and their target mRNAs involved in biological processes such as postsynaptic machinery, neurotransmission, and neuronal viability in AD [4]. Finally, the results of an approach to identify common signature patterns across public AD studies suggest that AD gene regulatory networks (GRNs) show significant enrichment for key signaling mechanisms involved in neurotransmission. Among the prioritized genes were prominent in synaptic transmission, and implicated in cognitive deficits [5].

Point 6: “Finally, I would like to ask why the authors choose the databases included in the manuscript and not others? What turns these more attractive? I noticed that these databases come from studies with small sample groups.”

Response 6: We truly appreciate this question. We choose these 3 databases after evaluating distinct microarray databases also present in Gene Expression Omnibus. We used as criterion selection that microarray expression data showed a normal distribution in the histogram, a similar median across all samples and that Alzheimer's disease and healthy samples grouped apart or with a tendency to separate in PCA plot (Please see lines 130 - 147).

Final note:

We highlighted the new changes in green in the revised version.

References

  1. Jha, S.K.; Jha, N.K.; Kumar, D.; Sharma, R.; Shrivastava, A.; Ambasta, R.K.; Kumar, P. Stress-Induced Synaptic Dysfunction and Neurotransmitter Release in Alzheimer's Disease: Can Neurotransmitters and Neuromodulators be Potential Therapeutic Targets? J Alzheimers Dis 2017, 57, 1017-1039, doi:10.3233/JAD-160623.
  2. Morton, H.; Kshirsagar, S.; Orlov, E.; Bunquin, L.E.; Sawant, N.; Boleng, L.; George, M.; Basu, T.; Ramasubramanian, B.; Pradeepkiran, J.A.; et al. Defective mitophagy and synaptic degeneration in Alzheimer's disease: Focus on aging, mitochondria and synapse. Free Radic Biol Med 2021, 172, 652-667, doi:10.1016/j.freeradbiomed.2021.07.013.
  3. Sadleir, K.R.; Kandalepas, P.C.; Buggia-Prevot, V.; Nicholson, D.A.; Thinakaran, G.; Vassar, R. Presynaptic dystrophic neurites surrounding amyloid plaques are sites of microtubule disruption, BACE1 elevation, and increased Abeta generation in Alzheimer's disease. Acta Neuropathol 2016, 132, 235-256, doi:10.1007/s00401-016-1558-9.
  4. Ceylan, H. Integrated Bioinformatics Analysis to Identify Alternative Therapeutic Targets for Alzheimer's Disease: Insights from a Synaptic Machinery Perspective. J Mol Neurosci 2022, 72, 273-286, doi:10.1007/s12031-021-01893-9.
  5. Kawalia, S.B.; Raschka, T.; Naz, M.; de Matos Simoes, R.; Senger, P.; Hofmann-Apitius, M. Analytical Strategy to Prioritize Alzheimer's Disease Candidate Genes in Gene Regulatory Networks Using Public Expression Data. J Alzheimers Dis 2017, 59, 1237-1254, doi:10.3233/JAD-170011.

Round 2

Reviewer 1 Report

The authors corrected the manuscript according to suggestions.  Corrected manuscript may be accepted for publication.